# Renal Cell Carcinoma in Tuberous Sclerosis Complex

**DOI:** 10.3390/genes12101585

**Published:** 2021-10-08

**Authors:** Elizabeth P. Henske, Kristine M. Cornejo, Chin-Lee Wu

**Affiliations:** 1Pulmonary and Critical Care Medicine, Brigham and Women’s Hospital, Harvard Medical School, Boston, MA 02115, USA; 2Department of Pathology, Massachusetts General Hospital, Harvard Medical School, Boston, MA 02114, USA; kcornejo@mgh.harvard.edu (K.M.C.); cwu2@mgh.harvard.edu (C.-L.W.)

**Keywords:** tuberous sclerosis complex, renal cell carcinoma, eosinophilic solid cystic RCC, RCC with leiomyomatous stroma, TSC1, TSC2, hybrid oncocytic chromophobe tumor, chromophobe RCC

## Abstract

Tuberous sclerosis complex (TSC) is an autosomal dominant disorder in which renal manifestations are prominent. There are three major renal lesions in TSC: angiomyolipomas, cysts, and renal cell carcinoma (RCC). Major recent advances have revolutionized our understanding of TSC-associated RCC, including two series that together include more than 100 TSC-RCC cases, demonstrating a mean age at onset of about 36 years, tumors in children as young as 7, and a striking 2:1 female predominance. These series also provide the first detailed understanding of the pathologic features of these distinctive tumors, which include chromophobe-like features and eosinophilia, with some of the tumors unclassified. This pathologic heterogeneity is distinctive and reminiscent of the pathologic heterogeneity in Birt–Hogg–Dube-associated RCC, which also includes chromophobe-like tumors. Additional advances include the identification of sporadic counterpart tumors that carry somatic *TSC1/TSC2/mTOR* mutations. These include unclassified eosinophilic tumors, eosinophilic solid cystic RCC (ESC-RCC), and RCC with leiomyomatous stroma (RCCLMS). A variety of epithelial renal neoplasms have been identified both in patients with tuberous sclerosis complex (TSC) and in the nonsyndromic setting associated with somatic mutations in the *TSC1* and *TSC2* genes. Interestingly, whether tumors are related to a germline or somatic *TSC1/2* mutation, these tumors often display similar morphologic and immunophenotypic features. Finally, recent work has identified molecular links between TSC and BHD-associated tumors, involving the TFEB/TFE3 transcription factors.

## 1. Tuberous Sclerosis Complex: Genetic and Clinical Features

### 1.1. TSC Overview

TSC is caused by germline loss-of-function mutations of either the *TSC1* or *TSC2* gene. TSC affects multiple organs, including the brain (subependymal giant cell astrocytomas (SEGAs), cerebral cortical tubers), heart (rhabdomyomas), kidney (angiomyolipomas (AMLs), cysts, renal cell carcinomas), lung (lymphangioleiomyomatosis (LAM)), and skin (angiofibromas). TSC patients can also develop seizures, autism, and cognitive disability. Approximately 60% of germline TSC gene mutations are de novo and 40% are inherited. Tumors in TSC, including angiomyolipomas and renal cell carcinomas, develop after somatic “second hit” inactivation of the remaining wild-type allele of *TSC1* or *TSC2*. The TSC complex integrates signals from the cellular environment, including growth factors and nutrients, to regulate the activity of the mechanistic target of rapamycin complex 1 (*mTORC1*). *mTORC1* controls numerous essential metabolic processes, including protein and lipid synthesis, glycolysis, ATP production, lysosomal biogenesis, mitochondrial function and biogenesis, and autophagy (to name just a few). In cells with biallelic inactivation of either *TSC1* or *TSC2*, *mTORC1* is hyperactive, leading to increased cell growth, tumorigenesis, and extensive metabolic reprogramming. Pivotal clinical trials have demonstrated that the *mTORC1* inhibitors sirolimus (rapamycin) and everolimus (Afinitor) partially decrease the size of AML and SEGA [1,2] and prevent loss of lung function in LAM [3,4]. When mTORC1 inhibitory therapy is stopped, the AML and SEGA tend to regrow and lung function decline resumes. Therefore, continuous (and perhaps lifelong) therapy is required to suppress disease.

### 1.2. Renal Manifestations of TSC

Renal disease is a major source of morbidity and mortality in TSC [5,6,7] and was the leading cause of TSC-associated death in a series of 284 TSC patients in the UK [8]. Renal disease in TSC usually begins in childhood. A recent analysis of 80 children in Belgium highlights the burden of renal disease; among 80 children with TSC, with median age of 0.8 years at first presentation and median f/u of 10.2 years, 75% had renal manifestations and 7.5% developed end-stage renal disease during the time period of the study [9].

The renal manifestations of TSC, which have been reviewed elsewhere [6,10,11,12,13,14], include angiomyolipomas, epithelial cysts which can resemble polycystic kidney disease (PKD), and RCC. Angiomyolipomas are benign tumors with three distinct components: vessels (often tortuous and dilated, with thickened asymmetric walls), abnormal smooth muscle-like cells that express melanocytic proteins, and adipose cells. Interestingly, genetic analyses have revealed all three of these cellular elements arise from a common precursor cell [15]. Renal cysts are common in TSC, with onset often in childhood, and usually have no clinical consequences. However, some individuals with TSC have germline contiguous deletion of *TSC2* and the adjacent *PKD1* gene on chromosome 16p13. This contiguous gene syndrome can lead to early-onset polycystic kidney disease (sometimes evident at birth, with abdominal distention from kidney cystogenesis) and a high risk of renal insufficiency. This contiguous gene syndrome was critical to the positional cloning of the *TSC2* gene [10,16]. The third renal manifestation of TSC and the focus of this review is TSC-associated RCC (TSC-RCC). We used the search terms tuberous sclerosis complex and renal cell carcinoma to identify publications relevant to this topic in PubMed. We include RCC that occurs in individuals with TSC as well as sporadic tumors with *TSC1* or *TSC2* gene mutations.

## 2. TSC-RCC, Clinical and Pathologic Features

The first series of TSC-RCC, consisting of six patients, was published in 1996 [13]. This small study revealed an early age at onset compared with sporadic RCC (36 years in TSC vs. 60–65 years in the general population), a female predominance (with RCC in the general population having a male predominance), and variable pathologic features. Nearly 20 years after this first series, two analyses of TSC-RCC, one from the Cleveland Clinic and the other from Massachusetts General Hospital, have allowed the histologic and clinical features of TSC-RCC to be systematically defined for the first time (Table 1).

Guo et al. [17] studied 57 RCC from 18 TSC patients with a striking female predominance (13 female and 5 male). The mean age at the time of surgery was 42 years (range 7–65 years). Nine patients had multiple tumors (ranging from 2–20). No distant metastatic disease had occurred in the 15 patients for whom follow-up was available. Three major morphologies were observed. The predominant morphology (59%) was “chromophobe-like”, although the authors note that in some cases, this pattern could have also been classified as hybrid oncocytic/chromophobe tumor (HOCT). The second most common pattern was renal angiomyoadenomatous tumor (RAT) or RCC with smooth muscle stroma. These tumors had nests of large cells with abundant clear cytoplasm surrounded by thick bundles of smooth muscle cells. The third most common morphology was unclassified, with eosinophilic tumor cells that formed both solid foci and cysts. Importantly, none of the tumors was HMB-45-positive, clearly distinguishing them from angiomyolipoma. Yang et al. [18] studied 46 RCCs from 19 TSC patients, also with a marked female predominance (12 female, 6 male). The mean age was 30 years (range from 7–59 years). No distant metastasis had occurred in the 14 patients for whom follow-up was available. The predominant histologic pattern was termed TSC-associated papillary RCC (PRCC). These tumors tended to have a branching papillary architecture, although some had a nested pattern. The tumor cells were large with clear cytoplasm; cells with finely granular, eosinophilic cytoplasm were also present in many of the tumors. The second most common pattern was HOCT, with tumor cells that had eosinophilic cytoplasm and centrally placed round nuclei (resembling oncocytoma) admixed with tumor cells exhibiting perinuclear cytoplasmic clearing (resembling ChRCC). The third pattern was unclassified, with eosinophilic cytoplasm.

**Table 1 genes-12-01585-t001:** Subtypes of TSC-associated renal cell carcinoma.

Authors	Types of RCC Seen in TSC Patients	# Type of Tumor/Total	%
Yang et al. [18]	TSC-associated papillary RCC	24/46	52
	Hybrid oncocytic chromophobe tumor	15/46	32
	Unclassifiable	7/46	15
Guo et al. [17]	Chromophobe	34/57	59
	Eosinophilic hybrid macrocystic	6/57	10
	RCC with smooth muscle stroma	17/57	29

#, Number.

Together, these studies have allowed TSC-RCCs to be placed into three major categories: TSC-associated papillary RCC, HOCT/chromophobe-like RCC, and unclassified/eosinophilic variants.

### 2.1. Tuberous Sclerosis Complex-Associated Papillary Renal Cell Carcinoma (TSC-Associated PRCC)

TSC-associated PRCCs are often encapsulated by a prominent fibromuscular stroma and composed of tumor cells with voluminous clear to eosinophilic cytoplasm forming complex branching architecture and less commonly display solid/nested or tubuloalveolar growth (Figure 1). The tumors typically exhibit features of either the International Society of Urology Pathology (ISUP)/World Health Organization (WHO) nuclear grade 2 (54%) or grade 3 (46%). Immunohistochemically, the tumor cells are positive for CK7, CAIX, and PAX8, with absence of staining for CD117 and SDHB [18]. Later, the absent or weak SDHB staining was felt to relate to the abundant clear cytoplasm typically identified in these neoplasms, which relates to the presence of intracellular mitochondria, accounting for a false negative immunohistochemical staining pattern and not a result of a true genetic defect in the succinate dehydrogenase gene [19]. This group of tumors were also described in a separate study as renal cell carcinoma with leiomyomatous stroma (RCC-LMS) also known as renal angiomyoadenomatous tumor (RAT), which encompasses a subset of RCCs composed of tumor cells displaying abundant clear cytoplasm with well-defined cell membranes of which the majority are of ISUP/WHO nuclear grade 2. The tumor cells are commonly arranged in solid nests and tubules and occasionally show papillary architecture. They contain thick bundles of smooth muscle within the neoplasm, often surrounding and encircling islands of tumor, resulting in a multinodular appearance. Immunohistochemically, the tumor cells are positive for CK7, CAIX, and PAX8, with absence of staining for CD117 [17]. 

Upon closer inspection, it appears TSC-associated PRCC and RCC-LMS/RAT neoplasms described in TSC show overlapping morphological and immunophenotypic features including abundant clear cytoplasm of tumor cells which show a predominantly tubulopapillary and solid growth pattern, the presence of a prominent fibromuscular stroma, and positivity for CK7 and CAIX without staining for CD117. Interestingly, only the tumors that display this morphology in patients with TSC have been found to metastasize, including a case in which images were not provided but described as displaying clear cell features [13,17,18]. 

### 2.2. Hybrid Oncocytic/Chromophobe Tumor (HOCT); Chromophobe-Like RCC

HOCTs are renal epithelial neoplasms that display morphologic characteristics of both an oncocytoma and chromophobe RCC (ChRCC) in varying amounts (Figure 2). Tumors described as HOCT display two main histologic patterns; the first comprises polygonal tumor cells with abundant eosinophilic cytoplasm and round centrally located nuclei resembling an oncocytoma but exhibiting perinuclear halos or clearing, similar to ChRCC. The second mosaic pattern shows an admixing of areas with features characteristic of ChRCC and oncocytoma. Immunohistochemically, the tumor cells are positive for CD117 and typically immunoreactive for CK7 in the majority of cases, with absence of staining for CAIX [18]. 

The tumors described as chromophobe-like RCC display sheets and nests of round cells with eosinophilic cytoplasm, irregular nuclear membranes, binucleation, and perinuclear halos, of which the majority resemble the eosinophilic variant of ChRCC. Areas that showed oncocytoma-like nuclear features were also noted in all cases. Immunohistochemically, the tumors are positive for CK7 in the majority of cases, with CD117 staining in a small subset (17%) and absence of staining for CAIX [17]. 

### 2.3. Unclassified Renal Cell Carcinoma (Unclassified RCC); Granular Eosinophilic Macrocystic RCC

The tumors described as having granular eosinophilic macrocystic morphology display cysts of varying sizes admixed with solid nests of tumor cells with abundant granular eosinophilic cytoplasm, round nuclei, and prominent nucleoli (Figure 3). The cysts are often lined by a single layer of tumor cells, frequently with a hobnail morphology. Immunohistochemically, these tumors show variable CK7 staining and are negative for CAIX and CD117 [17,20]. 

A subset of tumors that were labeled as unclassified RCC similarly display solid nests of tumor cells with voluminous eosinophilic and granular cytoplasm associated with cysts of varying sizes which show a hobnail appearance. Immunohistochemically, these tumors predominantly show an absence of staining for CK7, CAIX, and CD117 [18]. 

Tumors with this morphology were later termed eosinophilic solid cystic RCC (ESC-RCC) (see below).

### 2.4. Conclusions from Two Large TSC-RCC Series

What are the take-home messages from these two series that together examined 103 cases of TSC-RCC?

First, these series put to rest any concern that RCC in TSC represents a misdiagnosis (for example, AML mimicking RCC) since all of the tumors in these two series were HMB-45-negative.

Second, these series confirm the finding of the original smaller series that TSC-RCC occurs at a much earlier age than sporadic RCC and affects children and young adults. The median age at onset of RCC in the combined series was 36 years, and tumors occurred in children as young as age 7 in both series.

Third, these series confirm that TSC-RCC has a 2:1 female predominance; this contrasts with the general population, in which RCC has a male predominance.

Fourth, RCC in TSC has a heterogeneous histologic appearance, with at least three distinct morphologies (Table 1). This represents a striking contrast to other hereditary syndromes, including von Hippel–Lindau disease, SDH-associated RCC, and hereditary papillary RCC, in which a single histology predominates. Interestingly, RCC in BHD also exhibits heterogeneous histology, with HOCT as a predominant histology, highlighting potential parallels in the pathogenesis that will be discussed later. An improved understanding of how these tumors relate to rare and/or unclassified sporadic RCC is important, as discussed below.

## 3. Genetic Features of TSC-RCC

The *TSC1* and *TSC2* genes behave as classical tumor suppressor genes, with germline loss-of-function mutations accompanied by somatic loss-of-function of the remaining wild-type allele in tumors including angiomyolipomas [10,21]. Loss of heterozygosity (LOH) for either *TSC1* or *TSC2* in TSC-RCC was first reported by Bjornsson et al. [13], with LOH identified in three of the five tumors analyzed. This LOH is consistent with the two-hit tumor suppressor mechanism that also underlies the development of angiomyolipomas [22,23]. 

Kwiatkowski et al. [24] performed whole exome sequencing on seven RCCs from two patients with germline *TSC2* mutations. Second hit events were found in six of the seven tumors (loss of heterozygosity in five and a *TSC2* missense mutation in the sixth), confirming the two-hit mechanism. The second hit events appeared to be distinct in different tumors from the same patient, consistent with a multifocal origin. Bah et al. [25] studied a family with a germline *TSC2* missense variant (p.Arg905Gln) and found evidence of genetic second hit events in 5/6 TSC-RCCs, again confirming that loss of both alleles of *TSC1* or *TSC2* underlies renal tumorigenesis in TSC.

Kwiatkowski et al. performed the first whole exome sequencing on TSC-RCC [24]. This analysis of seven tumors revealed no additional “driver” mutational events. This important result suggests that loss of both alleles of *TSC1* or *TSC2* is sufficient to drive RCC in TSC. This is intriguing since multiple additional genetic events drive tumor progression in clear cell RCC.

## 4. *TSC1* and *TSC2* Mutations in Sporadic RCC

The fact that TSC patients develop RCC leads to the obvious question: do TSC gene mutations occur in sporadic RCC?

### 4.1. TSC1 and TSC2 Mutations in Sporadic ccRCC and chRCC

*TSC1* and *TSC2* mutations occur in clear cell RCC, although at a low frequency (2–5% [26,27,28,29,30], Table 2). *TSC1* and *TSC2* mutations also occur at a low frequency in sporadic chromophobe RCC [31,32] (4–10%, Table 3).

### 4.2. TSC1 and TSC2 Mutations in Sporadic Unclassified Eosinophilic RCC

Unclassified RCC represents about 10% of all RCC. An important recent development has been the discovery that the TSC genes are mutated in novel eosinophilic subtypes of RCC, some of which were previously unclassified (Table 3). Chen et al. [35] studied seven patients with a mean age of 54 and morphologically distinct sporadic RCCs with eosinophilic features and vacuolated cytoplasm that were unclassified. *TSC2* inactivating mutations were discovered in three of the tumors, and mTOR activating mutations were discovered in two others, associated with increased phospho-S6 and phospho-4EBP1. Cathepsin K was positive in five of the seven tumors, but HMB45, MelanA, and TFE3 were negative. Tjota et al. performed genomic sequencing on 18 cases of eosinophilic RCC with unusual morphologic features and found loss-of-function *TSC1* mutations in 6 and *TSC2* mutations in 8 [38] (Table 2). In a second paper, Tjota et al. [37] analyzed eight more cases of RCC with abundant granular eosinophilic cytoplasm, originally diagnosed as ChRCC, and found mutations in *TSC1* in 1/8 (12%), in *TSC2* in 3/8 (37%), and in *mTOR* in 4/8 (50%) cases. Williamson et al. [39] studied an RCC that mimicked a translocation RCC, with mixed clear and eosinophilic cytoplasm, but was negative for TFE3 and TFEB translocation/amplification by in situ hybridization and a gene fusion assay. This tumor was found to have a truncating mutation in *TSC1*.

### 4.3. TSC1 and TSC2 Mutations in Sporadic RCC-LMS (RCC with Leiomyomatous Stroma)

RCCLMS is an emerging subtype of RCC in which tumor nodules are separated by a distinct mesenchymal smooth muscle stroma [19,42]. Shah et al. studied 23 RCCLMS cases [36]. The patients had a mean age of 52 years, with a 2:1 female predominance and an average tumor size of 2.3 cm. The tumor nodules were separated by smooth muscle stroma and were CK7-positive. Mutations in *TSC1* (4), *TSC2* (4), *mTOR* (6), or *ELOC* (2) were detected (Table 3). The authors conclude that these tumors represent the sporadic counterpart of morphologically identical tumors occurring in TSC patients.

### 4.4. TSC1 and TSC2 Mutations in Sporadic Eosinophilic Solid and Cystic Renal Cell Carcinoma (ESC-RCC)

These RCCs display solid and cystic architecture composed of tumor cells with round nuclei and voluminous eosinophilic and granular cytoplasm with cytoplasmic stippling, which represent aggregates of rough endoplasmic reticulum [43,44,45]. The cysts are typically lined by cells with hobnail arrangement. These tumors are morphologically identical to those previously described as granular eosinophilic macrocystic RCC that arise in patients with TSC [43,44,46]. 

In the sporadic setting, ESC-RCCs are found predominantly in females and are associated with biallelic loss or mutations of either the *TSC1* or *TSC2* gene [34,38,46,47,48] (Table 3). Mehra et al. studied seven ESC-RCCs and found biallelic loss of *TSC1* or *TSC2* in six of the seven cases, with evidence of mTOR activation by IHC, leading the authors to conclude that there is a pathognomic role for TSC mutations in ESC-RCC [47]. Palsgrove et al. [34] also found TSC1/2 mutations in 8/9 pediatric and 6/6 adult ESC-RCCs, and Munari et al. [48] found *TSC1/2* mutations in 3/3 sporadic ESC-RCCs (interestingly, all were *TSC1* mutations). They also report an ESC-RCC in a TSC patient.

ESC-RCCs are typically diagnosed at a low stage and exhibit indolent behavior, but there have been rare cases with adverse prognosis [34,43,44,46,49,50]. The few patients with reported metastatic disease all lacked clinical features of TSC [49,50,51]. Immunohistochemically, the tumor cells are positive for CK20, at least focally, in the majority of cases. [43,44,45,46]. Additionally, they are usually negative for CK7, CAIX, and CD117 [38,43,44,45]. 

### 4.5. TSC1 and TSC2 Mutations in Sporadic Hybrid Oncocytic/Chromophobe Tumor (HOCT) and Chromophobe-Like RCC

HOCTs are morphologically similar to the tumors described in patients with TSC; the eosinophilic renal tumors that fit into these categories were also found to harbor mutations in *TSC1*, *TSC2,* or *MTOR* by next-generation sequencing [38]. Immunohistochemically, these tumors displayed variable CK7 and CD117 staining but lacked positivity for CK20 in most tumors [38]. 

TSC1 and TSC2 mutations occur in sporadic high-grade oncocytic tumor (HOT) of the kidney and eosinophilic vacuolated tumor (EVT) of the kidney. The majority of these tumors are sporadic, arising in patients with no clinical features or family history of a syndromic association, with the exception of only one case identified in a TSC patient [35,45,52,53]. These encompass a group of nested or solid tumors composed predominantly of high-grade nuclei with oncocytic cells, often exhibiting very large nucleoli and striking intracytoplasmic vacuoles. The neoplastic cells lack the irregular raisinoid nuclei or perinuclear clearing typically seen in ChRCC and often display nuclear features equivalent to ISUP/WHO grade 3 [35,45,52]. Immunohistochemically, these tumors are frequently positive for CD117; cathepsin K, at least focally if not diffusely; and CD10, while usually negative or with rare staining for CK7 and CK20 [45,52]. Next-generation sequencing has identified somatic mutations in TSC2 and mTOR in a subset of cases [35,54]. All of these tumors thus far present with a low stage and appear to behave indolently [45]. 

### 4.6. TSC1 and TSC2 Mutations in Sporadic Low-Grade Oncocytic Tumor (LOT) of the Kidney

LOTs are often small, solitary sporadic tumors that are typically identified in older patients in a nonsyndromic setting [45,55]. More recently, multiple tumors have been identified in a patient with clinically silent TSC who was found to harbor a germline *TSC1* mutation [56]. This is a group of tumors that display overlapping morphologic features with ChRCC and oncocytoma but do not entirely fit into either of these entities. Morphologically, the tumor cells are composed of oncocytic cells with low-grade round to oval nuclei and perinuclear halos. Significantly irregular nuclear membranes or raisinoid nuclei are typically not seen [45]. Immunohistochemically, they have a characteristic profile showing diffuse positivity for CK7 and absence of staining for CD117 [45,55]. Recently, a group of tumors that correlate with LOT were found to harbor *TSC1/2* and *MTOR* mutations [37,56]. All of these tumors to date appear to behave indolently and present at a low stage [45,55]. Kapur et al. performed whole exome sequencing on six LOTs that were previously classified as chromophobe RCC, eosinophilic variant. One had a TSC1 mutation that was discovered to be a germline mutation. Four had mTOR mutations, and one had a Rheb mutation [41]. 

### 4.7. TSC1 and TSC2 Mutations in Eosinophilic Vacuolated Tumor (EVT)

Mihaela et al. sequenced 19 EVTs (also called high-grade oncocytic tumors) and found nonoverlapping mutations in TSC1 (4), TSC2 (7), and mTOR (8) [40]. 

## 5. Pathogenesis of TSC-RCC: Links to TFEB/TFE3 and BHD

Recent work has identified a potential link between the TFEB/TFE3 transcriptional factors and TSC-RCC. The TFEB/TFE3 transcription factors are master regulators of lysosomal biogenesis via binding to “CLEAR” elements in the promoters of lysosomal genes [57]. mTORC1 phosphorylates TFEB and TFE3, leading to 14-3-3 binding and cytoplasmic sequestration [58]. TSC-RCCs express high levels of lysosomal proteins, including NPC1, and surprisingly have nuclear localization of TFEB, despite high mTORC1 activity [59]. In cellular models of TSC, TFEB and TFE3 are also found to be nuclear-localized (where they drive the expression of lysosomal genes and proteins), despite high levels of mTORC1 activity. Interestingly, TSC-RCCs were once described as TFE mimics. If TFE3/TFEB are drivers of TSC-RCC, this might explain some of the histologic phenotypes of TSC-RCC, including the eosinophilia, since TFE3/TFEB promote mitochondrial biogenesis [60]. Mouse models also support a link between TSC-RCC and BHD-RCC. As noted earlier, HOCTs occur in both TSC and BHD, suggesting a pathogenic link, but the mechanisms are unknown since TSC and FLCN have opposing effects on mTORC1 [61,62]. Recent data suggest that this “missing link” between BHD and TSC may involve TFEB. In a model in which FLCN is deleted in mouse kidney using the KSP-cadherin promoter, resulting in markedly enlarged cystic kidneys, deletion of TFEB almost completely normalizes kidney size and histology [63]. 

## 6. Conclusions

In conclusion, recent work has clarified the histologic features of TSC-associated RCC, based on more than 100 tumors in two large series. In parallel, sporadic RCCs with loss of function mutations in *TSC1* or *TSC2* have been identified, with some histologies such as eosinophilic solid cystic (ESC) and RCC with leiomyomatous stroma (RCCLS) showing high-frequency *TSC1* or *TSC2* mutations, and ESC-RCC is believed to be potentially pathognomonic for TSC gene mutations. The recent discovery that the TFEB/TFE3 transcription factors are hyperactive in both TSC and BHD provides a molecular link between TSC-associated RCC and BHD-associated RCC, both of which can include hybrid oncocytic chromophobe tumors. Critical next steps include the identification of targeted therapeutic strategies for individuals with unresectable or metastatic forms of these rarer subtypes of RCC.

## Figures and Tables

**Figure 1 genes-12-01585-f001:**
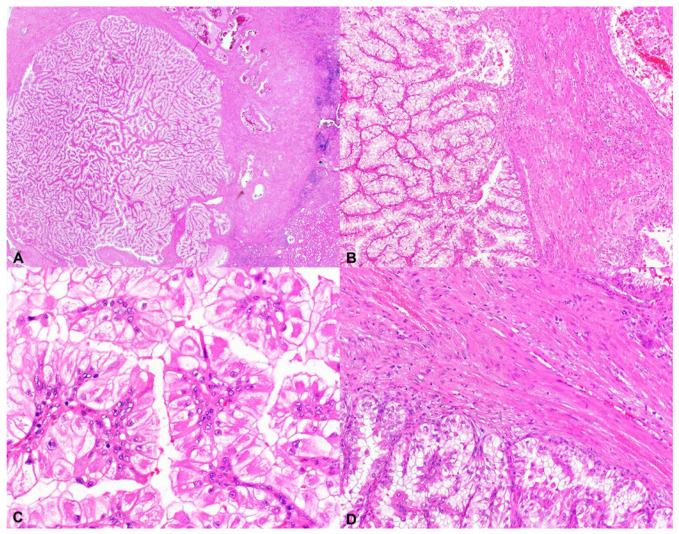
TSC-associated papillary RCC. (**A**–**D**) Low- and high-power views showing tumor cells with voluminous clear cytoplasm and complex branching architecture encapsulated by a prominent fibromuscular stroma (H&E 20×, 100×, 400×, 200×).

**Figure 2 genes-12-01585-f002:**
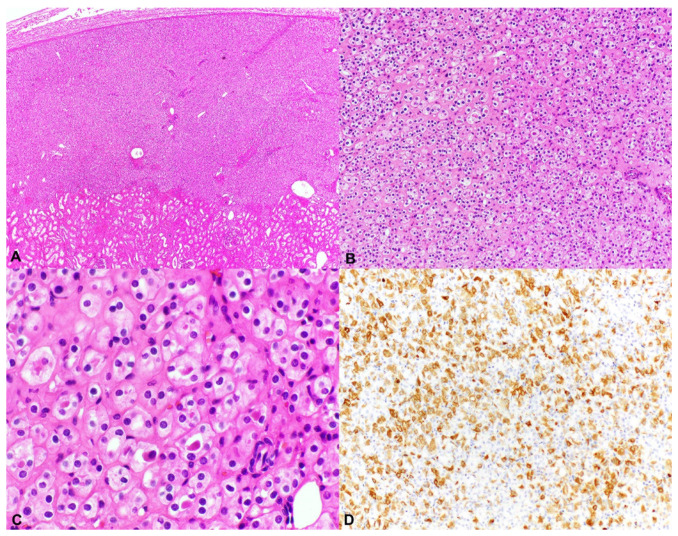
Hybrid oncocytic/chromophobe tumor (HOCT). (**A**–**C**) Low- and high-power views showing polygonal tumor cells with abundant eosinophilic cytoplasm and round centrally located nuclei resembling an oncocytoma but with perinuclear halos or clearing, seen in ChRCC (H&E 20×, 100×, 400×), and (**D**) characteristic positivity for CD117 (100×).

**Figure 3 genes-12-01585-f003:**
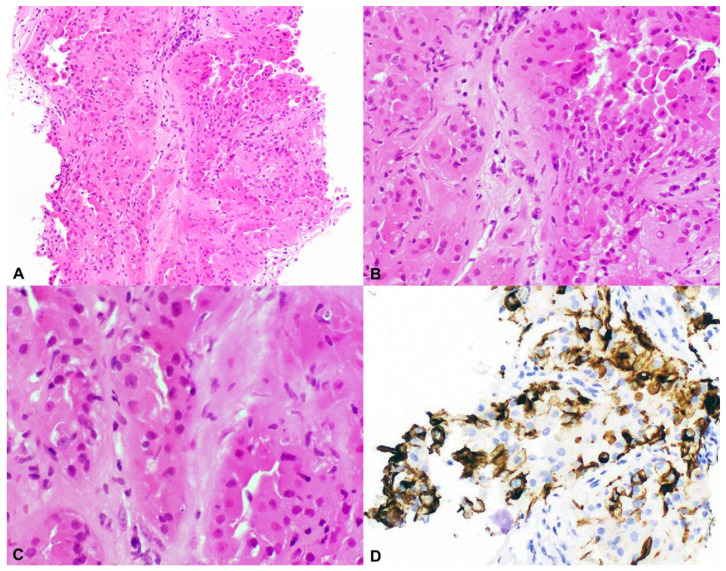
Eosinophilic solid and cystic RCC (ESC-RCC). (**A**–**C**) Low- and high-power views showing a solid and cystic tumor with granular eosinophilic cytoplasm, round nuclei, and prominent nucleoli (H&E 40×, 100×, 200×). (**D**) Tumor cells typically display CK20 positivity (100×).

**Table 2 genes-12-01585-t002:** TSC1, TSC2, and mTOR mutations in sporadic clear cell carcinoma.

Authors, Journal, Year	Histology	#	TSC1 Mutations # (%)	TSC2 Mutations # (%)	mTOR Mutations # (%)
Davis et al., *Cancer Cell*, 2014 [32]	ccRCC	448	4 (0.9%)	6 (1.2%)	32 (7%)
Sato et al., *Nat Gen*, 2013 [27]	ccRCC	106	2 (2%)	2 (2%)	6 (6%)
Kucejova et al., *Mol Cancer Res*, 2011 [26]	ccRCC	77	4 (5%)	0 (0%)	not done
Scelo et al., *Nat Comm*, 2014 [33]	ccRCC	94	not done	not done	8 (8.5%)

Abbreviation: ccRCC, clear cell renal cell carcinoma; #, number.

**Table 3 genes-12-01585-t003:** TSC1, TSC2, and mTOR in other subtypes of renal cell carcinoma.

Authors, Journal, Year	Histology	#	TSC1 Mutations # (%)	TSC2 Mutations # (%)	mTOR Mutations # (%)
Davis et al., *Cancer Cell*, 2014 [32]	ChRCC	66	3 (5%)	3 (5%)	2 (3%)
Durinck et al., *Nat Gen*, 2015 [30]	ChRCC	49	1 (2%)	1 (2%)	2 (4%)
Casuscelli et al., *JCI Insight*, 2017 [29]	ChRCC	79	2 (2.5%)	4 (5%)	4(5%)
Bah et al., *J Pathol Clin Res*, 2018 [25]	RCCLS	5	0	5 (100%)	not done
Palsgrove et al., *Am J Surg Pathol*, 2018 [34]	ESC-RCC	15	6 (40%)	8 (53%)	not done
	Unclassified ESC-like	3	1 (33%)	1 (33%)	not done
	Oncocytoid RCC	1	0%	1 (100%)	not done
Chen et al., *Am J Surg Pathol*, 2019 [35]	ESC-RCC	7	0	4 (42%)	2 (28%)
Shah et al., *Am J Surg Pathol*, 2020 [36]	RCCLS	18	4 (22%)	4 (22%)	6 (33%)
Tjota et al., *Hum Pathol*, 2020 [37]	ChRCC, eosinophilic variant	6	1 (16%)	2 (33%)	3 (50%)
	ChRCC	2	0	1(50%)	1 (50%)
Tjota et al., *Am J Surg Pathol*, 2020 [38]	ChRCC, eosinophilic variant	4	1 (25%)	2 (50%)	1(25%)
	Unclassified RCC	8	1 (12%)	5 (62%)	1(12%)
	CHRCC	3	2 (66%)	1 (33%)	0
	Renal oncocytoma	2	2 (100%)	0	0
Williamson et al., *Genes Chromosome Cancer*, 2020 [39]	ESC-RCC or HOCT	1	1 (100%)	0	0
Mihaela et al., *Mod Pathology*, 2021 [40]	Eosinophilic vacuolated tumor (EVT)	19	4 (21%)	7 (36%)	0
Kapur et al., *Mod Pathology*, 2021 [41]	Low-grade oncocytic tumor (LOT)	21	1 (4%)	0	7 (33%)

Abbreviation: RCCLS, renal cell carcinoma with leiomyomatous stroma; ChRCC, chromophobe renal cell carcinoma; ESC-RCC, eosinophilic solid and cystic RCC; HOCT, hybrid oncocytic/chromophobe tumor; #, number.

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
