# Peer review of "Renal Cell Carcinoma in Tuberous Sclerosis Complex"

_genes, 2021, doi:10.3390/genes12101585_

Round 1
Reviewer 1 Report
The manuscript by Henske et al is a review of the pathologic features of Tuberous Sclerosis Complex-Associated Renal Cell Carcinoma based in published series of cases of patients. In one, article published in 2014, they participated as authors. They define three subtypes, with morphologic and immunohistochemical characteristics. Tuberous Sclerosis Complex is an autosomal dominant disorder due to heterozygous germline mutations of TSC1 or TSC2. Somatic mutations of TSC1 and TSC2 have been observed in sporadic renal cell carcinoma, with morphology similar to that of renal cell carcinoma Tuberous Sclerosis Complex indicating a common molecular mechanism. Pathological diagnosis is essential to identify hereditary forms of cancer.
Major topics:
Bibliographic search strategies with inclusion and exclusion criteria should be included.
The text lacks a structure that would make it easier to read. I suggest the authors develop a table with the subtypes of renal cancer associated with germline and somatic mutations in TSC genes.
I also suggest the authors to create a figure with the molecular pathways involved in the pathogenesis of renal cancer.
Minor topics:
References 29, 41 and 45 are incomplete.
Tables 1 and 2 are untitled.
Author Response
Major topics:
- Bibliographic search strategies with inclusion and exclusion criteria should be included.
Response 1: We have added a sentence about the search strategies. “We used the search terms tuberous sclerosis complex and renal cell carcinoma to identify papers relevant to this topic.”
- The text lacks a structure that would make it easier to read. I suggest the authors develop a table with the subtypes of renal cancer associated with germline and somatic mutations in TSC genes.
Response 2: We have organized the text by subtype of tumor. We added a new Table (Table 1) with the major subtypes of RCC seen in TSC patients with germline mutations. Tables 2 and 3 contain the subtypes associated with somatic TSC gene mutations. We have added two new papers to Table 3 that were published since the Review was submitted.
- I also suggest the authors to create a figure with the molecular pathways involved in the pathogenesis of renal cancer.
Response 3: The precise molecular pathways involved in the pathogenesis of these rare forms of RCC are not yet well defined, and we believe that it is premature to try to create a figure at this stage. The hypothesis that TFEB plays an important role is discussed in the final section.
Minor topics:
- References 29, 41 and 45 are incomplete.
Response 1: We thank the reviewer for noticing the missing page numbers. Reference 45 and the two new references added (54 and 55) are online ahead of print. Hence, there are no page numbers. References 29 and 41 have been cited as suggested by the respective journals and page numbers were added.
- Tables 1 and 2 are untitled.
Response 2: titles were added
Reviewer 2 Report
1) General comments
The authors reported TSC related RCC from both histologic aspect and genetic aspect. This study must be informative for clinicians.
I am certain that this paper is well written and provides useful information for patients with TSC related RCC.
Author Response
General comments
The authors reported TSC related RCC from both histologic aspect and genetic aspect. This study must be informative for clinicians.
I am certain that this paper is well written and provides useful information for patients with TSC related RCC.
Response: We thank the reviewer for these positive comments.